

# Increased copy number of syncytin-1 in the trophectoderm is associated with implantation of the blastocyst

Luyan Guo[1,*], Fang Gu[2,*], Yan Xu[2] and Canquan Zhou[2,3]

[1] Department of Obstetrics and Gynecology, Sun Yat-Sen University First Affiliated Hospital, Guangzhou, Guangdong, China
[2] Reproductive Medical Center, Department of Obstetrics and Gynecology, Sun Yat-Sen University First Affiliated Hospital, Guangzhou, Guangdong, China
[3] Key Laboratory of Reproductive Medicine of Guangdong Province, Guangzhou, Guangdong, China
[*] These authors contributed equally to this work.

## ABSTRACT

**Background**. A key step in embryo implantation is the adhesion to and invasion of the endometrium by the blastocyst trophectoderm. The envelope proteins of HERV-W and -FRD (human endogenous retrovirus-W and -FRD), syncytin-1 and syncytin-2, are mainly distributed in the placenta, and play important roles in the development of the placenta. The placenta originates from the trophectoderm of the blastocyst. It is unclear whether the envelope proteins of HERV-W and -FRD have an effect on the development of the trophectoderm and whether they have any association with the implantation of the blastocyst.

**Methods**. The whole-genome amplification products of the human blastocyst trophectoderm were used to measure the copy number of syncytin-1 and syncytin-2 using real time qPCR. In addition, clinical data associated with the outcome of pregnancies was collected, and included age, body mass index (BMI), basic follicle stimulating hormone(bFSH), rate of primary infertility and oligo-astheno-teratospermia, the thickness of the endometrium on the day of endometrial transformation, the levels of estrogen and progestin on the transfer day, the days and the morphological scores of the blastocysts. The expression of mRNA and the copy numbers of syncytin-1 and syncytin-2 in H1 stem cells, and in differentiated H1 cells, induced by BMP4, were measured using real time qPCR.

**Results**. The relative copy number of syncytin-1 in the pregnant group (median: 424%, quartile: 232%–463%, $p < 0.05$) was significantly higher than in the non-pregnant group (median: 100%, quartile: 81%–163%). There was a correlation ($r_s = 0.681$, $p < 0.001$) between the copy number of syncytin-1 and blastocyst implantation after embryo transfer. As the stem cells differentiated, the expression of NANOG mRNA decreased, and the expression of caudal type homeobox 2(CDX2) and $\beta$-human chorionic gonadotropin ($\beta$-hCG) mRNAs increased. Compared to the undifferentiated cells, the relative expression of the syncytin-1 mRNA was 1.63 (quartile: 0.59–6.37, $p > 0.05$), 3.36 (quartile: 0.85–14.80, $p > 0.05$), 10.85 (quartile: 3.39–24.46, $p < 0.05$) and 67.81 (quartile: 54.07–85.48, $p < 0.05$) on day 1, 3, 5 and 7, respectively, after the differentiation. The relative expression of syncytin-2 was 5.34 (quartile: 4.50–10.30),

Corresponding author
Canquan Zhou,
zhoucanquan@mail.sysu.edu.cn

7.90 (quartile: 2.46–14.01), 57.44 (quartile: 38.35–103.87) and 344.76 (quartile: 267.72–440.10) on day 1, 3, 5 and 7, respectively, after the differentiation ($p < 0.05$). The copy number of syncytin-1 increased significantly during differentiation.

**Conclusion**. Preceding the transfer of frozen embryos, the increased copy number of syncytin-1 in the blastocyst trophectoderm was associated with good outcomes of pregnancies.

# INTRODUCTION

Failure of the embryo to implant is a crucial limiting factor for early pregnancy and assisted reproduction (*Bashiri, Halper & Orvieto, 2018*). The determinants of implantation include the embryo, the receptivity of the endometrium, the fetal-maternal crosstalk, immunity, and other factors (*Ashary, Tiwar & Modi, 2018*). A key step during embryo implantation is the adhesion to and invasion of the endometrium by the blastocyst trophectoderm. In the process, various genes and associated molecules take part in modulating the intricate crosstalk between the embryo and the endometrium (*Achache & Revel, 2006*). Molecular biological investigation of the blastocyst trophectoderm has the potential of becoming a useful predictive tool for identifying the embryos with the best chance of implantion and successful pregnancy (*Ntostis et al., 2019*).

Human endogenous retroviruses (HERVs) are derived from infectious retroviruses that, millions of years ago, integrated into the human germ-line DNA (*Bannert & Kurth, 2006*). It is estimated that residue gene sequences of HERVs represent up to 8% of the human genome (*Lander, 2011*) and the sequences can be vertically transmitted to offspring in a Mendelian fashion (*Bannert & Kurth, 2006*). The viral RNA is reverse-transcribed into a double stranded DNA (dsDNA), which is commonly called a provirus and competent for the subsequent integration into the host cell genome. A classical proviral structure includes the main retroviral genes (gag, pro, pol, and env), flanked by the two long terminal repeats(LTRs) (*Chen, Foroozesh & Qin, 2019*). Most HERVs have accumulated numerous stop codons and frame shift mutations or have undergone recombination between their LTRs, leading to the loss of the entire internal region and leaving only a solo LTR. Most contemporary HERV sequences are unable to encode functional proteins (*Stoye, 2001*). But some families or family members of HERVs are naturally expressed in particular physiological contexts and display physiological functions. For example, syncytin-1 and syncytin-2 are mainly distributed in the placenta (*Soygur & Sati, 2016*). Syncytin-1, the envelope protein of HERV-W, mediates the fusion of placental cytotrophoblastic cells to multinucleated syncytiotrophoblast and differentiation of syncytium during placental development (*Mi et al., 2000*). Syncytin-1 is expressed throughout gestation (*Okahara et al., 2004*) in syncytiotrophoblast cells (*Kudaka et al., 2008*; *Malassine et al., 2005*). It is a target gene of the GCM1 transcription factor, which is itself downstream of cAMP-regulated protein kinase A (*Kner et al., 2005*; *Yu et al., 2002*). The envelope protein strongly

induced syncytia formation by interacting with the type D mammalian retrovirus receptor (hASCT2, human sodium-dependent neutral amino acid transporter type 2) (*Blond et al., 2000*). Its gene harbored a 5′ LTR functional promoter, exhibiting several binding sites for transcriptional regulators involved in the control of proliferation and differentiation (*Prudhomme, Oriol & Mallet, 2004*; *Frendo et al., 2003*). Syncytin-2, the envelope protein of human endogenous retrovirus-FRD (HERV-FRD), is fusogenic (*Blaise et al., 2003*) and immunosuppressive (*Mangeney et al., 2007*) in the placenta. It is expressed in villous cytotrophoblast cells, specifically lining the membranes of a subset of cells bordering the syncytiotrophoblast, but not in the syncytiotrophoblast itself (*Kudaka et al., 2008*; *Malassine et al., 2007*). Syncytin-2 has an immunosuppressive domain not found in syncytin-1 that may play a role in protecting the fetus from the maternal immune system (*Blaise et al., 2003*). Previous research on syncytin-1 and syncytin-2 mainly focused on the placenta and rarely involved the early development of the embryo. Recently, Bikem Soygur indicated that syncytin expression is a prerequisite for embryo implantation and placentation. Syncytin-1 is mainly expressed in the trophectoderm cells underlying the inner cell mass of the blastocyst (*Soygur & Moore, 2016*). Syncytin might be a marker indicating good implantation potential for early embryos.

The preimplantation genetic diagnoses (PGD) and screening (PGS) are techniques used for determining the genetic status of cells (usually single cell) that have been biopsied from oocytes/zygotes or embryos in assisted reproduction. The number of trophectoderm cells from a blastocyst biopsy is very low. Currently, multiple displacement amplification (MDA) generates abundant assay-ready DNA to perform broad panels of genetic assays through its ability to rapidly amplify genomes from single cell. The residual MDA products from the clinical measurements provided a chance to analyze the genomes of early embryos in the study. Previous studies found that HERV-W transcripts have the unique capacity to be mobilized by the LINE-1 (L1) retrotransposons. This occurs through the reverse transcription of RNA transcripts originating from preexisting HERV-W proviral insertions, and their subsequent integration into new chromosomal positions (*Ostertag & Kazazian, 2001*; *Cordaux & Batzer, 2009*; *Grandi & Tramontano, 2017*). Consequently, the copy number of the genes from the HERV-W family may be changed by the L1 machinery in a physiological and pathological context. Some researches suggested that multiple sclerosis patients have increased number of HERV-W copies (*Garcia-Montojo et al., 2013*), whereas schizophrenia or bipolar patients appear to have decreased numbers of such copies (*Perron et al., 2012*). But the changes in the copy number of syncytin-1 and syncytin-2 were unclear in the genomes of early embryos. The aim of the present study was to evaluate whether the copy numbers of syncytin-1 and syncytin-2 have an association with the outcome of the pregnancies of the blastocysts. Because of the damage to embryos by rebiopsies, the expressions of syncytin mRNA cannot be measured in embryos from the pregnant and non-pregnant groups. The in-vitro differentiation model of H1 stem cell line induced by BMP4 was used to mimic the formation of blastocyst trophectoderm. The copy number and mRNA expressions of syncytin-1 and syncytin-2 were measured before and after differentiation.

## MATERIALS AND METHODS

### The patient information

The patients received PGD or PGS treatments for single gene disorders or chromosome abnormalities at the First Affiliated Hospital of Sun Yat-sen University, from January 2016 to January 2018. After detections, the patients who had transferable embryos, i.e., embryos that did not carry mutated genes and had normal chromosomes, were included in the study. These patients each underwent a single frozen embryo replacement cycle and the outcome of their pregnancy was assessed during follow-up examinations. The inclusion criteria were premenopausal women aged 25–42, whose bFSH was 1-8IU/L and who had more than 10 bilateral sinus follicles. None of the patients had a history of hormone or antibiotic treatment for at least three months prior to undergoing the single frozen embryo replacement cycle. Women with endometriosis, polycystic ovarian syndrome, repeated implantation failures, recurrent pregnancy loss, heavy uterine bleeding, potential neoplasms, intrauterine adhesions, scar diverticulum, submucosal uterine fibroids or congenital uterine malformations were excluded from the present study. When a couple lived together and had a normal sex-life, without the use of contraception, if after 1 year the wife did not fall pregnant, the patient was diagnosed as infertile. If the patient had never been pregnant, the patient was diagnosed as having primary infertility. The diagnostic criteria for oligo-astheno-teratospermia was sperm concentration $<15 \times 10^6$/ml, progressive motility $<32\%$ and normal morphology $<4\%$. Our study was initiated after being approved by the Medical Ethics Committee of the First Affiliated Hospital of San Yat-sen University (# [2019]41). Waiver of Informed Consent was also approved by the Medical Ethics Committee of the First Affiliated Hospital of San Yat-sen University.

### The clinical samples

The genome-wide amplification products from 6 to 8 trophectoderm cells of the blastocysts were collected. The blastocysts of patients were not rebiopsied. The samples were residual products from clinical tests and one to three blastocysts from each patient were included in the study. The DNA samples were frozen at negative 80 °C.

### Clinical data collection

According to the outcomes of the pregnancies after the embryos were transferred, the two experimental groups were designated the pregnant and non-pregnant groups. The age, BMI, bFSH, the rate of primary infertility and oligo-astheno-teratospermia, the thickness of the endometrium on the day of the endometrial transformation, the levels of estrogen and progestin on the transfer day, the days and morphological scores of the blastocysts, the rate of biochemical pregnancy, the rate of singleton live birth and early miscarriage were collected. The grading by Gardner for scoring the morphology of the blastocysts was used (*Gardner & Schoolcraft, 1999*). If the score of a 5-day old blastocyst was 3AB, 3BB, 3BA, 3AA, 4AB, 4BA, 4BB, 4AA, 5AB, 5BB, 5BA or 5AA, it was classified as a quality embryo. However, when the score of a 6-day old blastocyst was 4AB, 4BB, 4BA, 5AB, 5BB, 5BA or 5AA, it was also classified as a quality embryo. The number of quality embryos in all of the blastocysts was collected. $\beta$-hCG measured in the urine and blood of patients were positive

on the 12th day after the transfer. In addition, 3 weeks after transfer, the gestation sac was found to be present in the intrauterine wall using ultrasound. We judged the patient to be pregnant and classified the embryo as being in the pregnant group. Biochemical pregnancy is defined as serum $\beta$-hCG levels >6 IU/L with no gestational sac (determined by ultrasound scan 3 weeks later), or declining serum $\beta$-hCG levels before the ultrasound scan. Early miscarriage was defined as an abortion before 12 weeks of the gestation.

## Cell cultures and stem cell differentiation

The human embryonic stem cell line H1 was purchased from American Type Culture Collection (ATCC). Prior to differentiation, H1 stem cell was cultured under feeder-free conditions on Geltrex (Thermo Scientific: A1413302, USA)-coated plates. The culture medium of H1 cell line was provided by a domestic manufacturer (Cellapy: CA1014500, Beijing, China), supplemented with 100 U/ml penicillin and 100 μg/ml streptomycin. Cells were passaged with 0.005M EDTA (Thermo Scientific: 15575020, USA) every 3–5 days.

Based on a previously described protocol (*Li et al., 2013*) with some modifications, H1 stem cell was first cultured for 2 days after the passage. Then, the stem cells were differentiated by using the Knockout DMEM/F12 medium (Thermo Scientific: 12660012, USA) with 20% serum alternatives (Thermo Scientific: A3181501, USA), 2mM/L L-glutamine, 0.1mM/L non-essential amino acid, 10ng/mL bone morphogenetic protein-4(BMP4, Thermo Scientific: PHC9534, USA), 100 U/ml penicillin and 100 μg/ml streptomycin, for a week. The medium was replaced every day.

## Phalloidin staining

Cells were fixed using 4% paraformaldehyde for 30 min at room temperature and immersed in phosphate buffered saline(PBS) with 0.1% Triton X-100(Sigma-Aldrich, St. Louis, MO) for 15 min. The cells were stained with actin-tracker green phalloidin (1:50, Beyotime, Beijing, China) at room temperature for 1 h. Cell nuclei were stained with 4′–6-Diamidino-2-phenylindole (DAPI, Beyotime, Beijing, China) for 10 min at room temperature. Images were captured using an inverted microscope (DMi8, Leica microsystems, Wetzlar, Germany) and the cross sectional area of cells was computed by ImageJ software.

## Quantitative real-time PCR analysis

The genomic DNA of 6–8 trophectoderm cells from each blastocyst was amplified with REPLI-g-Single Cell Kit (150343, Qiagen, Germany) according to the manufacturer's instructions. The amplification products were diluted 100 times with nuclease-free water and then used for detecting the changes in the copy number of the relevant genes. The genomic DNA of the undifferentiated H1 cells, and the differentiated H1 cells, were extracted on days 1, 3, 5 and 7 using the universal genomic DNA kit (CW2298S, CWBio, Beijing, China) according to the manufacturer's instructions. The genomic DNA was used for detecting the copy number. The specific primers for the syncytin-1, syncytin-2 and GAPDH used in the quantitative real-time PCR are listed in Table 1. For syncytin-1, syncytin-2 and GAPDH replication, the PCR cycle was 95 °C for 10 min, followed by 40

**Table 1  The primers for the syncytin-1, syncytin-2 and GAPDH used for the quantitative real-time PCR of genome DNA.**

| Gene | Primers |
| --- | --- |
| Syncytin-1 | 5′-atcataaatccccatggccctc- 3′ (forward) |
| | 5′-gacgctgcattctccatagaaac- 3′ (reverse) |
| Syncytin-2 | 5′-cggatgtcccttgatatttatacttgt- 3′ (forward) |
| | 5′-ctgacagagctaaggttctgattagtgt- 3′ (reverse) |
| GAPDH | 5′-ctgatgcccccatgttcgtc- 3′ (forward) |
| | 5′-caggggtgct aagcagttgg- 3′ (reverse) |

**Table 2  The primers of the NANOG, CDX2, hCG, syncytin-1, syncytin-2 and GAPDH used for the quantitative real-time PCR of mRNA.**

| Gene | The sequences of primers |
| --- | --- |
| NANOG | 5′-aactctccaacatcctgaacctc- 3′ (forward) |
| | 5′-gtcacaccattgctattcttcgg- 3′ (reverse) |
| CDX2 | 5′aaatccccctagtttcccaagac- 3′ (forward) |
| | 5′-gcaaagacagagaagagagtgga- 3′ (reverse) |
| hCG | 5′-caggactcctcttcctcaaagg- 3′ (forward) |
| | 5′-gaagcctttattgtgggaggatc- 3′ (reverse) |
| Syncytin-1 | 5′-cacacaaatagtctgcctaccct- 3′ (forward) |
| | 5′-tagatggtcataggggggcactaa- 3′ (reverse) |
| Syncytin-2 | 5′-catgccttaaaactcaaggagcc- 3′ (forward) |
| | 5′-gtctggggttacatagcctatgg- 3′ (reverse) |
| GAPDH | 5′-cagcctcaagatcatcagcaatg- 3′ (forward) |
| | 5′-catgagtccttccacgataccaa- 3′ (reverse) |

cycles of 95 °C for 15 s, 60 °C for 30 s, and 72 °C for 30 s. The copy number measurements of cell line were repeated four times.

Quantitative real-time RT-PCR analysis was used to measure the changes in the abundance of NANOG, CDX2, hCG, syncytin-1 and syncytin-2 mRNAs. The total RNA of the undifferentiated H1 cells and the differentiated H1 cells on days 1, 3, 5, and 7 were extracted using Trizol reagent (B511311-0100, Sangon, Shanghai, China) according to the manufacturer's instructions. Total RNA (1 μg) was used in a 20 μl first strand cDNA synthesis using a reverse transcription system (R222-01, Vazyme, Nanjing, China). The cDNA (2 μl) was used for the qPCR, and the specific primers used for NANOG, CDX2, hCG, syncytin-1, syncytin-2 and GAPDH are listed in Table 2. For quantifying NANOG, CDX2, hCG, syncytin-1, syncytin-2 and GAPDH, the qPCR cycle was 95 °C for 10 min, followed by 40 cycles of 95 °C for 15 s, 60 °C for 30 s, and 72 °C for 30 s. All the experiments were repeated four times.

## Statistical analysis

All analyses were conducted with SPSS 23.0 and GraphPad Prism 6.0 software. The results are presented as the mean ± SD for normally distributed data or as the median and interquartile range for skewed data. The differences between two groups were analyzed

**Table 3  The clinical data from patients of pregnant and non-pregnant groups.**

|  | Pregnancy group (n = 32 embryos) | Non-pregnancy group (n = 29 embryos) | p-value |
|---|---|---|---|
| Age (y) | 31.3 ± 3.9 | 31.4 ± 4.1 | 0.929 |
| BMI (kg/m$^2$) | 21.0 ± 2.4 | 20.7 ± 2.0 | 0.698 |
| Primary infertility (%) | 25.0% (8/32) | 22.2% (4/18) | 0.825 |
| oligo-astheno-terato-Spermia (%) | 46.9% (15/32) | 50.0% (9/18) | 0.832 |
| Basic FSH | 5.2 ± 1.0 | 5.4 ± 1.1 | 0.614 |
| Good blastocyst (%) | 96.9% (31) | 82.8% (24) | 0.156 |
| Thickness of endometrium (mm) | 9.0 ± 1.1 | 8.4 ± 1.6 | 0.109 |
| Estrogen (pg/ml) | 143.5 (116.0–204.8) | 129 (95.5–167) | 0.278 |
| Progestin (ng/ml) | 12.3 (8.5–16.4) | 12.3 (7.8–16.2) | 0.681 |
| Biochemical pregnancy | 0 | NA | NA |
| Singleton live birth (%) | 93.75% (30) | NA | NA |
| Early miscarriage (%) | 6.25% (2) | NA | NA |

**Notes.**
Values are presented as mean ± SE or median (interquartile range) or %(n).

by Student's t test for normally distributed data or by Mann–Whitney U test for skewed data. The differences among more than two groups were analyzed by Kruskal-Wallis H test. Categorical variables were expressed as percentages and analyzed with the chi-square test. Spearman's Rank Correlation Coefficient ($r_s$) was used to analyze the association between the copy number of syncytin-1 and blastocyst implantation. A $p$-value < 0.05 was considered to be statistically significant.

## RESULTS

### The clinical data of patients

Thirty-two embryos in the pregnant group and 29 embryos in the non-pregnant group were enrolled in the study. The woman in the two groups did not differ significantly in terms of their age, BMI, basic FSH, the incidence of primary infertility and oligo-astheno-teratospermia, the ratio of good blastocysts, the thickness of the endometrium on the conversion day, and the level of estrogen and progestin on the day of embryo transfer. These factors affecting the outcome of the pregnancy were balanced between the two groups. In addition, the rate of biochemical pregnancy, singleton live birth and early miscarriage in the pregnant group were also collected. The clinical characteristics of the patients in the study are listed in Table 3.

### The changes in the relative copy number of syncytin-1 and syncytin-2 between the pregnant and non-pregnant groups

The median of the copy number of syncytin-1 or syncytin-2 in the non-pregnant group was used as the references, respectively. The relative copy number of syncytin-1 in the pregnant group (median: 424%, quartile: 232%–463%) was statistically significantly higher ($p < 0.05$), than in the non-pregnant group (median: 100%, quartile: 81%–163%) (Fig. 1A). However, the difference in the relative copy number of syncytin-2 between the pregnant

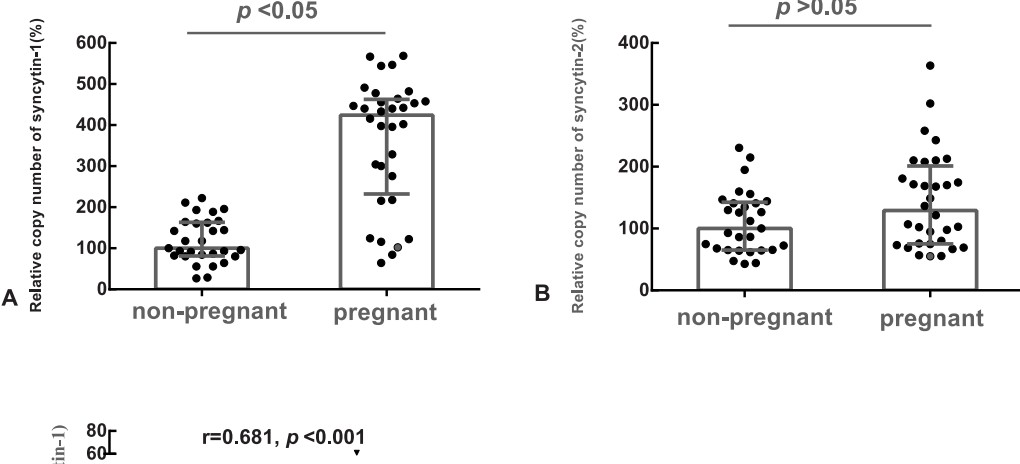

**Figure 1** **The relative copy number of syncytin-1 in the blastocyst trophectoderm was significantly associated with blastocyst implantation.** (A) The relative copy number of syncytin-1 between the non-pregnant and pregnant groups; (B) the relative copy number of syncytin-2 between the non-pregnant and pregnant groups; (C) Spearman Rank Correlation between the copy number of syncytin-1 and the outcomes of the pregnancies after embryo transfer.

group (median: 129%, quartile: 75%–201%) and non-pregnant group (median: 100%, quartile: 65%–142%) was not statistically significant ($p > 0.05$) (Fig. 1B).

## The copy number of syncytin-1 in the blastocyst trophectoderm was significantly associated with the outcomes of the pregnancy

Based on the balanced basic factors affecting the outcomes of the pregnancy between the pregnant and non-pregnant groups, the relation of the copy number of syncytin-1 and pregnancy outcome was analyzed using Spearman's Rank Correlation Coefficient ($r_s$). The copy number of syncytin-1 was significantly correlated ($r_s = 0.681$, $p < 0.001$) with the outcomes of the pregnancies. The implantation of the blastocyst is the first step for a successful pregnancy, the increased copy number of syncytin-1 in the blastocyst trophectoderm was significantly associated with blastocyst implantation (Fig. 1C).

## BMP4 induced differentiation of H1 stem cell line into trophoblast cells

The differentiation model of H1 stem cell line induced by BMP4 was used to mimic the formation of blastocyst trophectoderm. Before differentiation, the cytoplasm in the undifferentiated cells was little and the cells gathered together like a clone (Fig. 2). The average cross sectional area of undifferentiated H1 cells was $147.18 \pm 34.78 \, \mu m^2$ (Figs. 3A, 3B and 3C). After differentiation, the morphology of H1 cell line changed significantly.

The cytoplasm increased and the cell had increased protrusions and branches (Fig. 2). The average cross sectional area of differentiated H1 cells was $805.03 \pm 204.68\ \mu m2$ (Figs. 3D, 3E and 3F), which was larger than the area of undifferentiated H1 cells significantly (Fig. 3G, $p < 0.001$). The changes of NANOG, CDX2 and $\beta$-hCG mRNA were measured before and after differentiation of H1 cell line. The mRNA expression of undifferentiated cells was used as the reference. Embryo stem cells from the inner cell mass of the blastocyst have the potential of self-regeneration and multi-directional differentiation (*Edwards, 2004*). NANOG is a pivotal transcription factor maintaining the above potential of the stem cells. It is specifically expressed in undifferentiated stem cells and is not expressed in mature cells (*Chambers et al., 2003*). Compared to the undifferentiated cells, the relative expression medians of NANOG mRNA were 0.77 (quartile: 0.59–0.82, $p > 0.05$), 0.53 (quartile: 0.40–0.87, $p > 0.05$), 0.11 (quartile: 0.05–0.20, $p < 0.05$) and 0.02 (quartile: 0.01–0.04, $p < 0.05$) on day 1, 3, 5 and 7, respectively, after the differentiation (Fig. 4A). These showed that the pluripotency of H1 cell line had decreased. CDX2 plays a role from the blastocyst stage, by determining the formation of the trophectoderm (*Niwa et al., 2005*). As the differentiation proceeded, the relative expression medians of CDX2 mRNA were 1.44 (quartile: 1.37–1.73), 1.96 (quartile: 1.70–2.31), 2.94 (quartile: 2.53–4.27) and 1.88 (quartile: 1.59–2.04) on days 1, 3, 5 and 7, respectively, after the differentiation (Fig. 4B, $p < 0.05$). CDX2 is a marker of early trophoblast cells, the increased expression of which indicated that H1 cell line had differentiated into early trophoblast cells. The relative expression of $\beta$-hCG mRNA was transformed by the square root. Compared to the undifferentiated cells, the medians of square root of $\beta$-hCG mRNA expression were 1.37 (quartile: 0.93–1.94, $p > 0.05$), 1.60 (quartile: 1.08–2.43, $p > 0.05$), 3.84 (quartile: 2.41–5.27, $p < 0.05$) and 22.19 (quartile: 17.31–41.99, $p < 0.05$) on day 1, 3, 5 and 7, respectively, after the differentiation (Fig. 4C). $\beta$-hCG is a marker of mature trophoblasts, the increased expression of which indicated that H1 cell line had differentiated into trophoblast cells. The results suggested that H1 cell line was successfully induced to differentiate into trophoblast cells by BMP4.

## The expression of syncytin-1 and syncytin-2 mRNAs increased as H1 cell line differentiated into trophoblast cells

The mRNA expression of the undifferentiated cells was used as the reference. Compared to the undifferentiated cells, the relative expression medians of syncytin-1 mRNA were 1.63 (quartile: 0.59–6.37, $p > 0.05$), 3.36 (quartile: 0.85–14.80, $p > 0.05$), 10.85 (quartile: 3.39–24.46, $p < 0.05$) and 67.81 (quartile: 54.07–85.48, $p < 0.05$) on day 1, 3, 5 and 7, respectively, after the differentiation (Fig. 5A). The relative expression medians of syncytin-2 were 5.34 (quartile: 4.50–10.30), 7.90 (quartile: 2.46–14.01), 57.44 (quartile: 38.35–103.87) and 344.76 (quartile: 267.72–440.10) on day 1, 3, 5 and 7, respectively, after the differentiation ($p < 0.05$, Fig. 5B). As the differentiation proceeded, it was found that the expression of syncytin-1 and syncytin-2 mRNAs increased. The model of stem cell differentiation was used to simulate the differentiation of the trophectoderm from the blastocyst. The increased mRNA expression indicated that the syncytin-1 and syncytin-2 may play a role in the development of early trophoblasts.

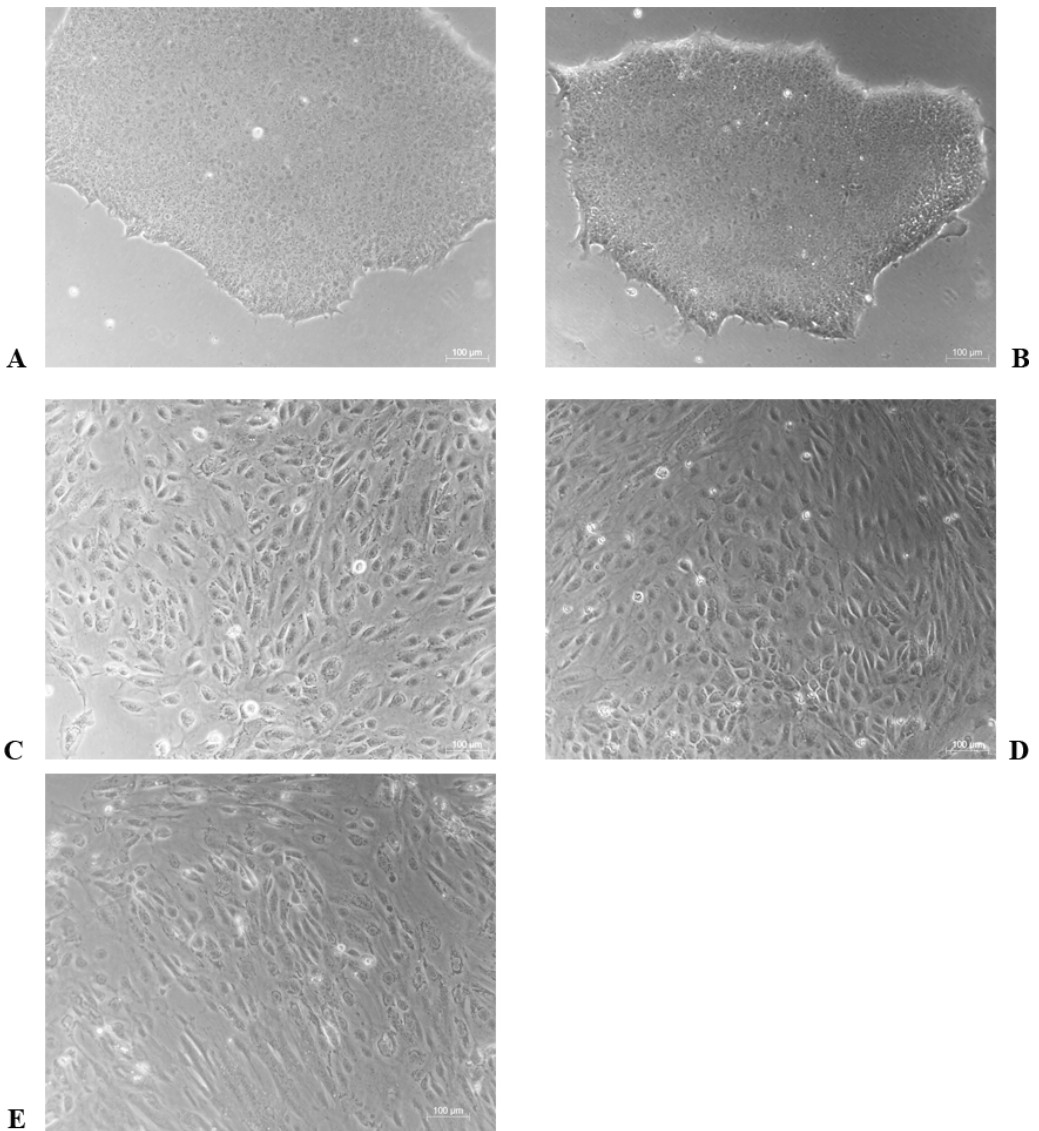

**Figure 2** **The morphological characteristics of the undifferentiated and the differentiated H1 cells.** (A) Undifferentiated H1 cells; (B) differentiated cells induced by BMP4 for 1 day; (C) differentiated cells induced by BMP4 for 3 days; (D) differentiated cells induced by BMP4 for 5 days; (E) differentiated cells induced by BMP4 for 7 days.

## The copy numbers of syncytin-1 increased as H1 cell line differentiated into trophoblast cells

The copy number of undifferentiated cells was used as the reference. Compared to the undifferentiated cells, the relative copy numbers of syncytin-1 were 1.55 (quartile: 1.51–1.71), 1.53 (quartile: 1.42–1.61), 1.55 (quartile: 1.39–1.79) and 1.82 (quartile: 1.50–2.01) on day 1, 3, 5 and 7, respectively, after differentiation ($p < 0.05$, Fig. 5C). The relative copy numbers of syncytin-2 were 1.13 (quartile: 0.93–1.23), 1.12 (quartile: 1.00–1.17), 1.16 (quartile: 1.11–1.54) and 1.18 (quartile: 1.07–1.33) on day 1, 3, 5 and 7, respectively,

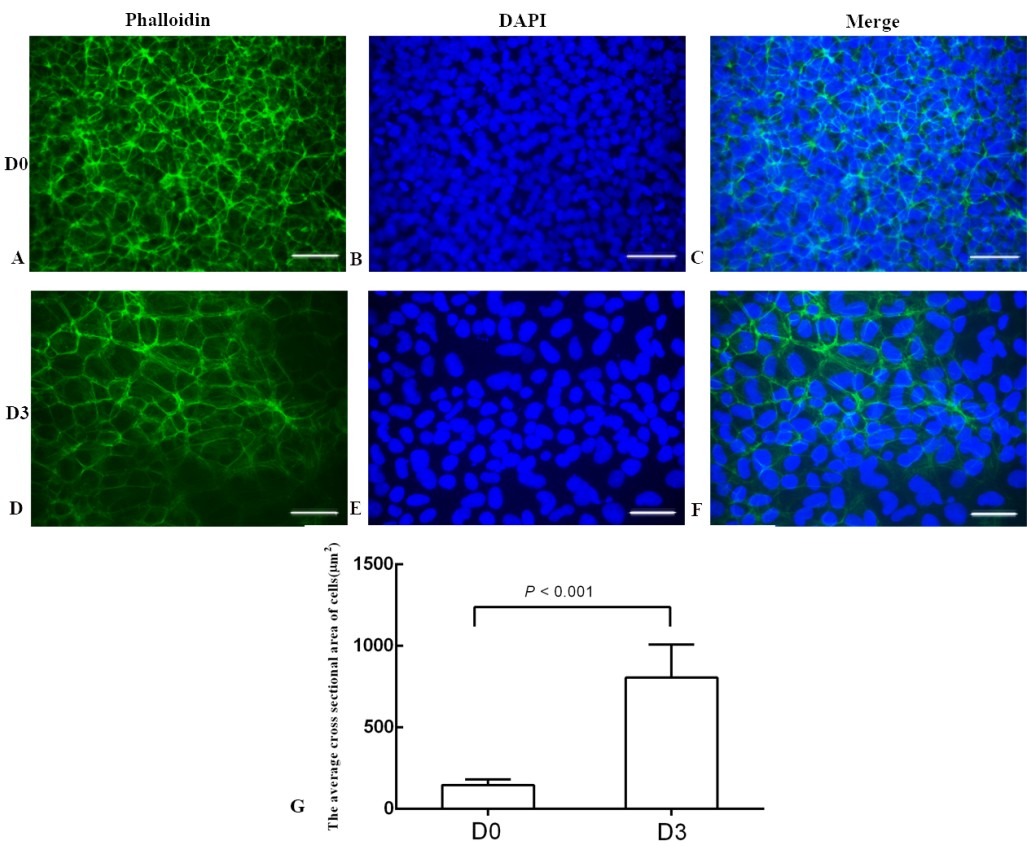

**Figure 3** **The phalloidin staining of the undifferentiated H1 cells and the differentiated H1 cells.** (A, B, C) The undifferentiated cells; (D, E, F) the differentiated cells on Day 3; (G) a bar graph showing the differences of the average cross sectional area of cells; the experiments were repeated three times; bar: 50 μm.

after the differentiation ($p > 0.05$, Fig. 5D). During the differentiation, the copy numbers of syncytin-1 increased significantly, but the copy number of syncytin-2 did not have significant change statistically.

## DISCUSSION

Improvements have been made in human assisted reproduction techniques (ART) over the past decades, but embryo implantation remains the rate-limiting step for the success of ART (*Spinella et al., 2018*). Although the PGD and PGS are used for screening potentially transferable embryos, i.e., embryos that do not carry any mutated genes and have normal chromosomes, despite this, not all of the transferable embryos implant successfully (*Sciorio, Tramontano & Catt, 2020*). Improving the implantation rate of embryos is an urgent issue for assisted reproduction. Previously, the clinicians mainly used the morphological grade of the embryos to prioritize the transferable ones (*Ziebe et al., 1997*). But high grade embryos, determined using the morphological grade, may not have good implantation potential. At present, researchers are putting more attention into determining how to identify the best

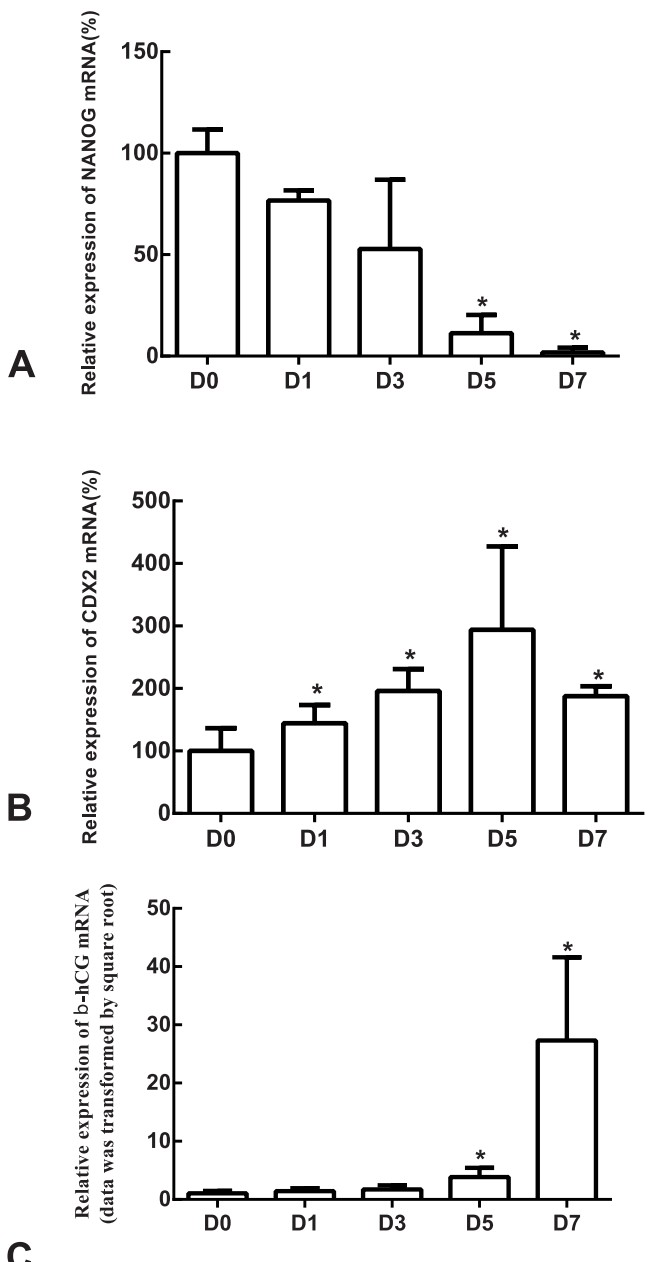

**Figure 4** **The expressions of NANOG, CDX2 and β-hCG mRNA in the undifferentiated H1 cells and the differentiated H1 cells on Days 1, 3, 5 and 7.** (A) The relative expression of NANOG mRNA; (B) the relative expression of CDX2 mRNA; (C) the relative expression of β-hCG mRNA; *indicates that compared with the undifferentiated H1 cells, the differences were statistically significant, $p < 0.05$; $n = 4$ for each group; CDX2, caudal type homeobox 2; β-hCG, β-human chorionic gonadotropin.

embryos that have a high potential for implantation and successful pregnancy (*Sigalos, Triantafyllidou & Vlahos, 2016*).

The attachment and implantation of the human blastocyst occurs via the polar TE near the inner cell mass(ICM) (*Gamage, Chamley & James, 2016*). Following implantation,

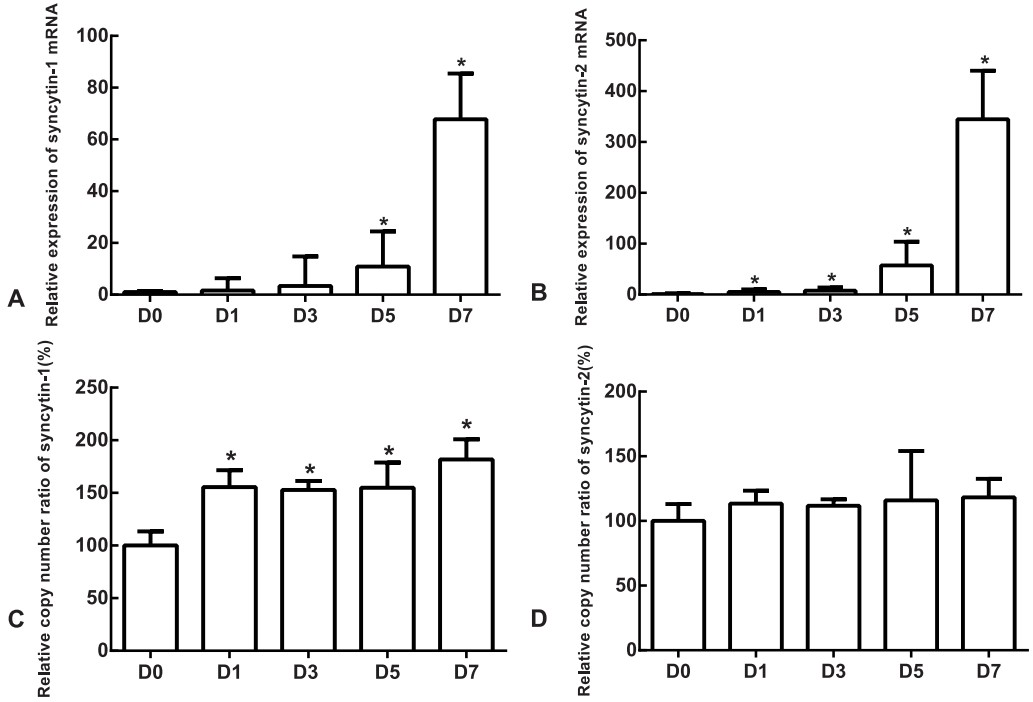

**Figure 5** **The expressions of mRNA and the copy numbers of syncytin-1 and syncytin-2 in the undifferentiated H1 cells and the differentiated H1 cells on Days 1, 3, 5 and 7.** (A) The relative expression of syncytin-1 mRNA; (B) the relative expression of syncytin-2 mRNA; (C) the copy number of syncytin-1; (D) the copy number of syncytin-2; *indicates that compared with the undifferentiated H1 cells, the differences were statistically significant, $p < 0.05$; $n = 4$ for each group.

human TE cells begin to differentiate into primitive mononuclear cytotrophoblast (CT) and primitive multinucleated syncytiotrophoblast (ST) (*Bos, Chamley & James, 2018*). It is known that syncytin-1 mediates the fusion of placental cytotrophoblastic cells to multinucleated syncytiotrophoblast and differentiation of syncytium during placental development (*Mi et al., 2000*), but its role in early embryo development remains unclear. A previous research showed that a number of HERVs were transcribed when the genome is first activated in around the 8-cell stage (*Grow et al., 2015*). Bikem et al. also showed that syncytin-1 protein expression was mainly in trophectoderm cells of human preimplantation blastocysts and it might have a function in embryo adhesion and attachment to endometrium. The HERV-W family is a multicopy, retrotransposably active family (*Belshaw et al., 2005*), with dozens of env-related regions in the human genome (*Voisset et al., 2000*). The copy number of the genes from the HERV-W family may be changed by the L1 machinery (*Ostertag & Kazazian, 2001*; *Cordaux & Batzer, 2009*; *Grandi & Tramontano, 2017*). In the present study, we found that compared with the non-pregnant group, the copy number of syncytin-1 from the blastocyst trophectoderm was significantly higher in the pregnant group. But the copy number of syncytin-2 in the pregnant group was similar to the copy number in the non-pregnant group. The regulatory mechanism of syncytin-2 expression might be different from syncytin-1. Syncytin-2 transcription was not

increased by changing the gene copy number at the DNA level. The transcription rate of the syncytin gene is also regulated by the promoter region in and close to the 5′ long terminal repeat region (5′ LTR) of the provirus (*Yu et al., 2002*).On the basis that the clinical factors affecting the outcomes of pregnancy were balanced between the two groups, we assessed the association between the copy number of syncytin-1 and the outcome of pregnancy using Spearman's Rank Correlation Coefficient. It was found that the copy number of syncytin-1 from the blastocyst trophectoderm was positively associated with pregnancy. This indicated that embryos with a higher copy number of syncytin-1 might have better developmental potential. The rebiopsy of trophectoderms of blastocysts will decrease the rate of survival and implantation of embryos of patients that have been given treatments of PGD or PGS (*Bradley et al., 2017*). To identify the association of syncytin copy number and gene expression, the model of stem cell differentiation was used. As the differentiation proceeded, it was found that the expression and copy number of syncytin-1 both increased. The expression of syncytin-2 mRNA increased, but its copy number did not change significantly at the DNA level. The increased copy number of syncytin-1 may provide more transcription templates to promote gene expression. The expression of syncytin-2 might be regulated by other mechanisms. To determine the regulatory mechanisms, more basic experiments are needed. The copy number changes were detected by using real time quantitative PCR, a method that has been used by many other researchers (*Providenti et al., 2006*; *Zhao et al., 2015*; *Paakkanen et al., 2012*). The primers in our experiment could specifically amplify the syncytin-1 and syncytin-2 genes in the genome (Table 1). The different quantity of primary templates from the biopsied blastocyst trophectoderm was corrected by using the synchronous measurement of GAPDH, as an internal reference gene. The measurement was suitable for the patients that received treatments of PGD and PGS as the residual amplification products from the biopsied trophectoderm could be reused. There are some differences in the gene expression of polar and mural trophectoderm (*Petropoulos et al., 2016*).The biopsy of blastocysts collected the mural trophectoderm. We analyzed the genome DNA of trophectoderm, which avoided the differences of gene expression between the polar and mural trophectoderm.

In the present study, there were still some limitations. The sample size of embryos was small and the embryos were all from patients that received PGD or PGS treatments for single gene disorders or chromosome abnormalities. Some patients suffered from primary infertility or oligo-astheno-terato-spermia. These factors will have some effects on the embryos, but they were balanced between the pregnant and non-pregnant groups. A prospective study on the copy number of syncytin-1 and pregnancy outcome among patients who have none of the above diseases and agree to biopsies of their embryos, would make a contribution to the conclusions of the present study.

## CONCLUSIONS

The increased copy number of syncytin-1 was associated with good outcomes of pregnancies after the transfer of frozen embryos. The expressions of syncytin-1 and syncytin-2 mRNA increased during the process of differentiation from stem cells into trophoblasts induced

by BMP4. The DNA copy number of syncytin-1 may be a potential marker in detecting embryos with high potential for implantation. The present study provides a new method for accurate selection of embryos to be transferred. The method is especially suitable for patients that have received PGD or PGS treatments. The clinicians could use the morphological grade of the embryos and copy number of syncytin-1 to prioritize the transferable embryos. This new method is expected to improve the success of ART.

### Funding

The study was supported by the National Key Research and Development Program of China (2016YFC1000205) and the Key Laboratory of Reproductive Medicine of Guangdong Province (2017B307030314164). The funders had no role in study design, data collection and analysis, decision to publish, or preparation of the manuscript.

### Grant Disclosures

The following grant information was disclosed by the authors:
National Key Research and Development Program of China: 2016YFC1000205.
Key Laboratory of Reproductive Medicine of Guangdong Province: 2017B307030314164.

### Competing Interests

The authors declare there are no competing interests.

### Author Contributions

- Luyan Guo conceived and designed the experiments, performed the experiments, analyzed the data, prepared figures and/or tables, authored or reviewed drafts of the paper, and approved the final draft.
- Fang Gu performed the experiments, analyzed the data, prepared figures and/or tables, authored or reviewed drafts of the paper, and approved the final draft.
- Yan Xu performed the experiments, authored or reviewed drafts of the paper, and approved the final draft.
- Canquan Zhou conceived and designed the experiments, authored or reviewed drafts of the paper, and approved the final draft.

### Human Ethics

The following information was supplied relating to ethical approvals (i.e., approving body and any reference numbers):

The Medical Ethics Committee of the First Affiliated Hospital of San Yat-sen University granted Ethical approval to carry out the study within its facilities (Ethical Application Ref:[2019]41).

### Data Availability

The raw measurements are available in the Supplementary Files.

## Supplemental Information

Supplemental information for this article can be found online at http://dx.doi.org/10.7717/peerj.10368#supplemental-information.

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
