# Peer review of "Increased copy number of syncytin-1 in the trophectoderm is associated with implantation of the blastocyst"

_PeerJ, doi:10.7717/peerj.10368_

## Round 0.1 · original submission · Major Revisions

Please fully address the reviewers' comments and return the revision back.

Reviewer 1 ·

Basic reporting

This is the review report for the manuscript titled ‘Increased copy number of syncytin-1 in the trophectoderm is associated with implantation of the blastocyst’ by Guo et al. Human trophoblast cells express syncytin-1 and syncytin-2 that are derived from HERV-W and HERV-FRD envelopes, respectively, and known to be expressed to form syncytiotrophoblasts in placenta. However, it is still unclear whether syncytin-1 and syncytin-2 have important roles during implantation. The authors showed syncytin-1 but not syncytin-2 has increased genomic copy number in the pregnant group when compared with the non-pregnant group even though there were no significant differences in the clinical conditions. The authors also conducted in vitro experiments using H1 embryonic stem cells to suggest that the increase of syncytin-1 but not syncytin-2 genomic copy number is associated with differentiation into trophoblast cells. The manuscript is concluded with these data that syncytin-1 could be the indicator for the successful implantation after frozen embryo transfer.
This study is quite interesting and important for the development of methods for accurate selection of embryos to be transferred. In particular, the clinical data of syncytin-1 copy number is valuable to gain insights into the strategy for embryo transfer. However, the manuscript needs to be improved before going to be published in the journal PeerJ.

Major comments
1. The English needs to be improved to make your text more clearly understandable. For examples, ‘We want to know… (lines 69-70)’, ‘The results indicated that… (lines 156-157)’, ‘As implantation of the … (lines 174-176)’, ‘The transcription factor CDX2…(lines 262-263)’can be proofread.

2. The Introduction is not effective to understand the background of this study. It is hard to understand how these syncytins are expressed and how these syncytins contribute to placental development. The authors should include more of what are known already about the roles and molecular characteristics of syncytin-1 and syncytin-2 during placentation to improve the Introduction.

Experimental design

3. The authors described that the aim of this study is to examine the association of genomic copy numbers and expression of syncytins with the outcome of the pregnancy (lines 69-70). This, especially the genomic copy number, gives off an erratic impression, the authors therefore should explain more details why they focused on the factor genomic copy number in this study.

4. The authors need to explain the purpose of the experiments using H1 cells before going ahead (line 178-).

Validity of the findings

5. The discussion about the differentiation from embryonic stem cells to trophoblast cells (lines 256-269) should be incorporated in the subsection ‘BMP4 induced differentiation of H1 stem cell line into trophoblast cells’ of Results to make it easy to understand the meanings of RT-qPCR data. This part (lines 256-269) also can be shrunk by excluding the redundant descriptions such as ‘We used BMP4…(lines259-260)’, ‘This indicated that the expression of NANOG… (lines 261-262).’, and ‘A previous study found that the trophectoderm… (lines 263-266)’.

6. Increasing the genomic copy number of syncytin-1 is the important finding of this study. Hence discussion about the phenomenon is necessary to increase the impact of this manuscript. I would suggest the authors to incorporate the explanation that how the genomic copy number affects gene expression as well as protein function (could be general). The authors should also discuss about why only syncytin-1 but not syncytin-2 altered genomic copy numbers.

Additional comments

1. Please explain what the A, B, C and D indicate in the figure legends of Figures 3 and 4.

2. In line 211, please delete ‘the’ before ‘the H1 cell line’

3. Delete ‘the’ before HERV (lines 58, 59, 62, 239)

4. Authors’ first and last names should not be separated by commas in the References.

·

Basic reporting

In overall, this manuscript is clear and data are interesting.
Concerning the scientific writing style, some small changes are required such as using adequate academic wording instead “so, we wanted” (line-19) and when necessary introducing passive voice sentences instead of the pronoun “we” (i.e. lines 196/200/213).
The abbreviation hCG for human chorionic gonadotropin is not the same throughout the manuscript and figures (i.e. lines 104/194 and table 2).
All abbreviations should be defined at their first use (line 63 – HERV-FRD, line 96- BMI) and redefined in each figure.
The flow in the material and methods is broken at a few places and requires some brief changes:
• line 7, “patients and sample collection” section is introducing DNA storage. I would suggest splitting into two sections the patient information and the clinical samples.
• lines 108- 111, “When a couple lived together and had a normal sex-life, without the use of contraception, if after 1 year the wife did not fall pregnant, the patient was diagnosed as infertile. If the patient had never been pregnant, the patient was diagnosed as having primary infertility” could be placed in “Patients and sample collection” section

Experimental design

The title “Increased copy number of syncytin-1 in the trophectoderm is associated with implantation of the blastocyst” suggests you looked at Implantation whereas you investigated until day 33. Could you develop in the discussion section your choice of date (biochemical vs clinical pregnancy in relation to Implantation)?
In your study, have you included couples suffering from unknown infertility or only male fertility factors?
The use of an in vitro model of stem cell differentiation to mimic in vivo embryonic trophoblast cell differentiation needs to be incorporated into the study aims section to contain the hypothesis.
The BMP4 induction protocol should be stated as a common practice and referenced.
The choice of statistical analysis t-tests for multiple comparisons seemed inappropriate and ANOVA / Mann Whitney tests are recommended. More details are required in the Methods section.
Lines 147 – 148: “Spearman’s Rank Correlation Coefficient (rs) was used to compare the relative odds of clinical factors associated with blastocyst implantation”. Could you develop how you measured odd for each embryo using Spearman’s rank? Could you demonstrate by using a correlation on a binary outcome that the Point-Biserial Correlation Coefficient is equal to Spearman’s rank in this study? How is the figure 1C different from the figure 1A?
The details about the number of technical and experimental replicates should be enumerated in each figure and present in the method section.

Validity of the findings

In your hypothesis, “whether the envelope proteins of HERV-W have an effect on the development of the trophectoderm”: it is either a cause or causality, but the data are limited and the mechanism is not fully established in your results. Therefore, the hypothesis should be rephrased.
Lines 178 – 180: “The small undifferentiated cells had obvious nuclei and gathered together like a clone. After differentiation, the morphology of the H1 cell line changed significantly. The cell volume was bigger and the cell had increased protrusions and branches (Figure 2)”. The research standards are not achieved with the wording “obvious” in the absence of a nuclear marker and the absence of quantification for the cell volume. In order to conclude these findings, a phase contrast-nuclei marker staining and measures of cell volume (using Phalloidin staining?) with statistical analysis are expected.
In Table 3, data for a few parameters are incomplete/missing such as the incidence of primary infertility (18 out of 29) or embryo quality (24 out of 29). However, it was stated in the line 104 “The number of quality embryos in all of the blastocysts was collected”.
The clinical collection finished in 2018. Could you update the manuscript and add an analysis of live birth/late miscarriages and syncytin DNA copies?
The discussion section requires to be developed on the study limitations (numbers, infertility, use of stem cells model, in vivo/in vitro…).
After identifying an increase in mRNA, the dataset would be strengthened by the identification of an increase at the protein level (i.e. syncytin-1) in the in-vitro model overtime. Therefore, additional work to investigate the protein levels are highly recommended.
In the conclusion section, line-282 “Syncytin-1 may be an important marker in detecting embryos”. Would you not conclude on the DNA copy number of Syncytin-1 as a potential marker instead of the functional protein?

Additional comments

The authors have identified an increase in DNA copy of syncytin-1 in differentiating trophectoderm in vitro and observed that a significantly higher number of copies was found in trophectoderm from embryos that implanted and are responsible for successful pregnancy until 33rd day compared to embryos in the “non-pregnant” group. It is a very interesting finding.
However, the manuscript requires some improvements: re-structuring and adding more details. Moreover, the key findings would be strengthened with some additional work.

·

Basic reporting

In this well written paper the authors investigate whether the fusogenic envelope proteins of HERV-W (human endogenous retrovirus-W), syncytin-1 and syncytin-2, are expressed by the trophoectoderm of the blastocyst and if they have any association with the implantation of the blastocyst.

The HERV-W proteins are mainly distributed in the placenta and play a key role in the fusion of the progenitor cytotrophoblast to the fused multinucleated syncytiotrophoblast. A key step in embryo implantation is the adhesion to and invasion of the endometrium by the blastocyst trophectoderm.


This is a paper with good novelty and with clinical significance as good markers of preimplantation success for a healthy pregnancy outcome are lacking in the IVF field. The patient groups are large and the ethics approval of the study and the statistical tests used are appropriate.

Experimental design

In this study the authors utilise whole-genome amplification of 6-8 cells of the human blastocyst trophectoderm to measure the copy number of syncytin-1 and syncytin-2 using real time qPCR. In addition, clinical data associated with the outcome of pregnancies was collected, including
age, BMI, bFSH, rate of primary infertility, the thickness of the endometrium on the day of
endometrial transformation, the levels of estrogen and progestin on the transfer day, the days and the morphological scores of the blastocysts. The authors also examine the expression of mRNA and the copy numbers of syncytin-1 and syncytin-2 in H1 stem cells, and in differentiated H1 cells, induced by BMP4, were measured using real time qPCR.

They show that the relative copy number of syncytin-1 in the pregnant group of patients was significantly higher than in the non-pregnant group following IVF with a single blastocyst. This was shown to significantly correlate (rs = 0.681, p < 0.001) with successful blastocyst implantation after embryo transfer. Syncytin 2 did not reach significance.

As expected as the H1 ES cells differentiated under BMP4 stimulation, shown by the expression of NANOG mRNA decreasing, and the expression of CDX2 and β-HCG mRNAs increasing asses by qRTPCR. The copy number of syncytin-1 increased significantly during differentiation.

The authors conclude that : Preceding the transfer of frozen embryos, the increased copy number of syncytin-1 in the blastocyst trophectoderm was associated with good outcomes of pregnancies.

Validity of the findings

Q1) The adhesion, attachment and infiltration of the blastocyst is mediated by the first syncytiotrophoblast originating from the fusion of the trophoectoderm layer. The authors should state this as it is directly relevant to the function of syncytin 1 and 2 in trophoblast fusion. It could be mentioned in the discussion in the context of increased syncytin 1 and a successful IVF outcome being dependant on the ability of the blastocyst to implant.

Additional comments

Q2) the photos of the H1 cells don’t really add anything, H1 are known to default to syncytiotrophoblast under BMP4 stimulation. The paper would be strengthened by doing IHC for syncytiotrophoblast markers such as CK7, AlkP, E –Cad.

Q3) Suggest the clinical significance is addressed in the conclusion.

---

## Round 0.2 · accepted · Accept

I will accept your paper but you still have the chance to make some changes following the comment of one of the reviewers during the process of proofing.

Reviewer 1 ·

Basic reporting

This is the review report for the revised manuscript titled ‘Increased copy number of syncytin-1 in the trophectoderm is associated with implantation of the blastocyst’ by Guo et al. The manuscript is carefully revised and English was improved.

Experimental design

no comment

Validity of the findings

Major comments
The genomic copy number of syncytin-1 in differentiated H1 cells was only 1.5-fold (Figure 5), which was quite lower than that observed in pregnant embryos (4-5-fold higher than non-pregnant embryos) (Figure 1). Moreover, the copy number in H1 cells was seemed to be already plateau on D1 after differentiation even though the expression was continuously increased until D7 (70-fold). These data support the authors’ hypothesis that increase of syncytin-1 copy number is associated with the successful pregnant, however, it seems to be difficult to find the association between the increase of copy number and expression amount. The manuscript may get improved by discussing the other possible meanings of syncytin-1 transposition. e.g. affecting the relevant genes expression through epigenetic modification.

Minor comments
1. Lines 16-18 must be revised. Syncytin-2 is not the envelope protein of HERV-W but HERV-FRD. Please reconsider and modify the sentence.

2. ‘It did not increase the transcripts by changing…’ (Lines 333-334) needs to be modified to such as ‘Syncytin-2 transcription was not increased by changing…’.

·

Basic reporting

The manuscript is well-referenced and unambiguous with a clear hypothesis.

Experimental design

Matching the hypothesis, the experiment design has been refined to achieve rigorous scientific standards.

Validity of the findings

In this version of the manuscript, the impact and limitations of the results have been clearly stated and discussed.

Additional comments

A well-written resubmitted manuscript which incorporated all answers to previous reviewers' comments. Your findings set the scene for further scientific projects to improve clinical ART rates.

·

Basic reporting

Thankyou to the authors for their extensive revisions to the paper. The changes have significantly improved the readability of the manuscript. All reviewers concerns have been fully addressed.

Experimental design

No concerns

Validity of the findings

No concerns

Additional comments

See above